# Effects of Adherence to Once-Daily Treatment on Lung Function, Bronchial Hyperreactivity and Health Outcomes in Adolescents with Mild-to-Moderate Asthmoka: A Twelve-Month Survey

**DOI:** 10.3390/children9121854

**Published:** 2022-11-29

**Authors:** Roberto W. Dal Negro, Paola Turco

**Affiliations:** 1National Centre for Respiratory Pharmacoeconomics and Pharmacoepidemiology, 37124 Verona, Italy; 2Research & Clinical Governance, 37100 Verona, Italy

**Keywords:** asthma, adherence to treatment, adolescents

## Abstract

Adolescents with asthma are usually insufficiently adherent to regular inhalation treatments, thus limiting their effectiveness. The aim of this study is to investigate the role of adherence to single-inhaler long-acting LABA/ICS dry-powder combination o.d. in affecting lung function, bronchial hyperreactivity, and health outcomes over a twelve-month survey of a group of non-smoking adolescents with mild to moderate asthma. **Methods**: Age, gender, BMI and atopy, forced expiratory volume in 1 s (FEV^1^), maximum mid-expiratory flow (MMEF), and maximum expiratory flow at 25% of lung filling (MEF_25_) were assessed via a Boolean selection process from the institutional database at recruitment, as well as after 6 and 12 months, together with the incidence of exacerbation, school days that were taken off, GP and specialist visits, and systemic steroid and/or antibiotic courses. Adherence was checked monthly via a direct telephone call. Statistics were calculated with an ANOVA trend analysis, assuming *p* < 0.05. **Results**: Two well-matched sample groups of 54 subjects each were obtained. The mean annual adherence to treatment ranged from 48.2% doses ± 10.9 sd to 79.3% doses ± 8.8 sd (*p* < 0.001), regardless of age and gender. Only adolescents that adhered to the o.d. ICS/LABA inhalation regimen progressively achieved complete control of all lung function parameters (FEV^1^: 0.001; MMEF: *p* < 0.002; MEF_25_ < 0.001; <0.001), minimized their bronchial hyperreactivity (*p* < 0.001), and optimized all health outcomes (*p* < 0.001—*p* < 0.002) over the survey duration. **Discussion**: A good adherence to treatment is essential for asthma management, particularly in young patients. Factors that are totally independent of the complexity of the therapeutic regimen adopted (namely, only a once-daily inhalation in the present survey) probably represent the major reasons limiting the adolescents’ adherence. Cultural, educational, behavioral, and psychological factors are frequently involved, are difficult to control, and can present barriers to adolescents’ asthma management. Further studies aiming to deeply understand and possibly remove the reasons for such adolescents’ attitudes are needed, in cooperation with actions oriented in this direction by families, educators, and health professionals.

## 1. Introduction

Inhaled medications are as crucial in asthma management as adherence to the therapeutic strategy adopted. Adolescents with asthma commonly have suboptimal adherence to inhaled treatments, thus limiting the effectiveness of whatever therapeutic strategy they are following [1,2]. Today, as in the past, considering the still poor diagnostic attention given to childhood asthma [3,4], several factors can contribute to insufficient asthma management: the low perception and the low awareness of asthma risks [5,6,7]; their own and their parents’ anxiety, somatization, and hostility [8]; the fear of tarnishing their own social image due to their perception of asthma as a segregating disability among their friends [9,10], and their difficulty in assuming responsibility in self-management [11,12] play a critical role in several cases. Obviously, the psychological profile of asthma-suffering adolescents and the lack of appropriate family and medical support are detrimental factors affecting the sufferer’s attitude to adhering to long-term therapeutic strategies. As adherence tends in any case to decrease over time [13], the negative effects of sub-optimal adherence on outcomes can be further enhanced significantly, in the case of adolescents [14]. 

It has been reported that inappropriate adherence to both preventer and reliever drugs via inhalation represents a major factor that is able to affect asthma control, particularly in children and teenagers [15,16]. On the other hand, it has been known for many years that therapeutic strategies, based on simple action plans and requiring a low frequency of inhalations per day, can contribute to better adherence to asthma treatment. In general, fixed-dose LABA/ICS applied through single-inhaler combinations actually tend to improve patients’ compliance substantially [17,18,19,20,21,22]. Moreover, the use of long-acting inhalation drugs, characterized by twenty-four-hour efficacy and then requiring only one device actuation per day, is supposed to further contribute to adherence improvement [23]. These drugs are supposed to be much more effective in terms of quality of life and health outcomes, particularly in asthmatic adolescents, as in the case of those subjects usually showing the lowest degree of adherence. The aim of this study was to assess how much the different degrees of adherence to prescribed medications can affect lung function and health outcomes in non-smoking adolescents with mild to moderate asthma, treated with a single-inhaler long-acting ICS/LABA dry-powder combination o.d. over the course of a twelve-month survey. 

## 2. Materials and Methods

The study was an observational, retrospective analysis of asthmatic adolescents referred to the Lung Unit of the Specialist Medical Centre (CEMS), Verona, Italy, over the period from February 2018 to September 2019. Data were obtained from the institutional, UNI EN ISO 9001-2008 validated database, and classic Boolean algebraic formulas were used for subject selection [24]. The database contains general, historical, clinical, and health-economic data, complete lung function, and therapeutic information for each patient referred to the Centre. At present, more than 96,000 respiratory patients are included in the database, which is continuously growing, with new patients added daily; it is also updated upon every patient’s visit.

The basic selection criteria were adolescents: with mild to moderate asthma, of both genders; ranging from 12 to 18 years of age; non-smokers; with normal cognitive function and without any relevant comorbidity; who have been prescribed (and then presumably taking) fluticasone fumarate/vilanterol 90/22 mcg dry powder o.d., via Ellipta. This combination was chosen because the drug is suitable for prescribing to adolescents within the age range of 12–18 years in our own country. Exclusion criteria were: the presence of comorbidities that make inhalation difficult; the refusal of parents’ informed consent; an incomplete set of clinical and lung function data; prescriptions of ICS/LABA o.d. that were different from fluticasone fumarate/vilanterol. 

The variables of sex, age, BMI, and atopy, forced expiratory volume in 1 s (FEV^1^), maximum mid-expiratory flow (MMEF), and maximum expiratory flow at 25% of lung capacity (MEF_25_) were assessed at recruitment (the index date), and after 6 and 12 months, their values being reported as a percentage of predicted value. At the same time, the extent of bronchial reactivity to methacholine (MCh) was also assessed in all subjects and was reported as the MCh dose (in mcg) inducing a 20% drop from their FEV^1^ baseline value (PD_20_ FEV^1^). At each time point of the survey, the number of exacerbations, school days taken off, GP and specialist visits, and systemic steroid and/or antibiotic courses were also recorded from the database. At the index date, these health outcomes were calculated over the previous six months and were compared to those measured over the two semesters of the study (namely, at the 6th and the 12th month). 

As the inhaler device used is provided with a clearly visible dose counter and the device contains enough doses of the drugs for exactly thirty days of treatment, the adolescents’ adherence to the inhalation regimen was calculated at the end of every month via telephone calls, during which each adolescent (or one of the adolescent’s parents) had to communicate the number of remaining doses visible in his/her device to the interviewer. The compliance was then calculated as the percentage of skipped inhalation doses/month, which corresponded to skipping days of treatment over each period of the survey. As established some time ago, subjects who took <70% of the prescribed inhalation doses were considered “non-compliant” [25]. Stemming from this assumption, the sample consisted of two subgroups with the same number of subjects (that is, for every subject who adhered to the treatment that was selected, one non-adherent subject would enter the sample) to compare the two subgroups head-to-head. An analysis of variance (ANOVA) was used for comparing the trends of all variables over the study periods, and *p* < 0.05 was assumed as the limit of statistical significance. Regarding ethics, the study was approved by the R & CG Ethical Committee during the session that was officially held on 10 October 2017 (code: 02/RG02/2017). Even though the selection of subjects was conducted by automatic procedures run from the database, informed consent was requested from the adolescents’ parents because monthly direct telephone contact had been planned for during the survey.

## 3. Results

According to the inclusion and exclusion criteria, a sample of 108 adolescents with mild to moderate atopic asthma was selected (see the selection flow in Figure 1). The general characteristics of the whole sample are reported in Table 1. 

Subjects were well-matched according to sex, and all subjects were atopic, with a clear prevalence of combined seasonal and perennial allergens. At recruitment, the mean values of the lung function indices showed the presence of mild to moderate obstruction, in particular, obstructions involving the peripheral airways. The bronchial reactivity to MCh also showed a mild to moderate response, independent of age and sex. The health outcomes recorded at the index date have also been reported in Table 1: although the general impact seems mild at first glance, it should be considered that all subjects were presumably already taking ICS/LABA regularly, even if it was to a variable extent.

The sample consisted of two subgroups, comprising 54 adherent and 54 non-adherent subjects. The mean annual adherence in the adherent group of adolescents was 79.3% doses ± 8.8 sd, while the adherence was 48.2% doses ± 10.9 sd in the non-adherent group (*p* < 0.001), without significant differences according to age and gender. Their corresponding trends of lung function variables are reported in Table 2, together with the significance levels of the corresponding statistical comparisons. It is clearly apparent that adolescents taking >70% of the prescribed doses (namely, those adhering to the regimen) progressively achieved complete control of all their lung function parameters (FEV^1^ *p* < 0.002; MMEF *p* < 0.002; MEF_25_ *p* < 0.001, respectively) and minimized the degree of their bronchial hyperreactivity (*p* < 0.001) over the survey duration. Conversely, all lung function parameters, and bronchial reactivity did not change significantly over the study in non-adherent adolescents, such as in those taking <70% of the prescribed doses (all *p* = ns).

Moreover, all health outcomes considered in the study proved the same trends (Table 3). In other words, starting from the first semester, the incidence of exacerbations systematically dropped (*p* < 0.002) over the survey duration, together with that of school absenteeism (*p* < 0.001), of GP and specialist visits (*p* < 0.001 and <0.002, respectively), and of systemic steroid and antibiotic courses (both *p* < 0.001) in the sub-group of subjects with good adherence, while it remained unchanged in those with low adherence (all *p* = ns).

Lung function, bronchial reactivity and health outcomes showed a progressive trend of improvement on a semester-by-semester basis, but this was only in those adolescents with good adherence to treatment.

## 4. Discussion

The effectiveness of therapeutic strategies against bronchial asthma is usually mainly investigated in terms of the drugs’ efficacy and tolerability. Nevertheless, even if contributing substantially to asthma morbidity per se, the role of non-adherence to treatment is much less frequently investigated, particularly in young patients, as in those subjects who are known to be less prone to coping with their persistent respiratory disorder and to compliance with the therapeutic strategies prescribed [26,27,28].

Several factors can contribute to the low adherence of asthmatic adolescents: cultural, educational, behavioral, and psychological variables are very difficult to control in many cases and frequently act as barriers to asthma management. Adherence has been generally reported to be very poor in adolescents, ranging from 25 to 35% in various studies, resulting in poor health outcomes [29,30,31]. Similar data were confirmed in a large study conducted on more than 70,000 adolescents with asthma: their adherence to twice-daily ICS/LABA resulted in 34% compliance at six months and 24% at twelve months [32]. It is of note that therapeutic strategies based on more than two inhalations per day and/or through different devices proved to contribute negatively in terms of adherence [33]. 

The strategy based on the once-daily administration of long-acting inhalation drugs has been supposed to further foster the patients’ adherence during long-term therapeutic strategies; several studies tend to confirm this hypothesis [15,34,35,36].

Although not aimed at calculating the prevalence of non-compliance in young asthmatics, data from the present study confirm that adherence is generally poor in adolescents with mild to moderate asthma and that all long-term outcome results are unchanged over the survey duration in these cases. Conversely, a good level of long-term adherence to the prescribed regimen proved to be quite effective and enabled the progressive improvement of lung function, bronchial hyperreactivity, and health outcomes. It should be emphasized that the significant poor adherence observed in the present study was recorded even though the therapeutic regimen prescribed to the adolescents was very simple (once daily), probably not interfering significantly with their daily activities. As is different from the results of some previous studies [37,38], the lack of a significant age- and gender-dependent difference found in the present survey was likely due to the peculiar study design of the survey and the narrow range of adolescent ages under consideration. 

The main message emerging from the study seems to be that factors completely independent of the complexity of the therapeutic regimen adopted (namely, treatment only once daily, in the present survey) can represent the major reasons affecting the adolescents’ levels of adherence to inhalation treatments. As mentioned above, too low an awareness of asthma [5,6,7], a psychological profile characterized by anxiety, depression, and hostility [8,39], the fear of being limited in personal and social activities, together with the perception of asthma as a disability, leading to a form of segregation from their friends [9,10], and finally, the difficulties in assuming any responsibility regarding self-management [11,12] likely play a dominant role in these cases. 

Obviously, the role of health professionals, families, and schools would be of great value in supporting these fragile personalities [40]. Otherwise, the impact of asthma will not be modified effectively in adolescents, regardless of the respiratory drug, inhalation device, and therapeutic strategy that is prescribed and adopted. Even if of mild to moderate severity, adolescent asthma will remain effectively uncontrolled when not properly managed. As a consequence, these adolescents’ distrust, anxiety, irritation, fear, and depression will increase, thus creating a dangerous and vicious circle, leading to the progressive increase of asthma’s social and economic burden.

As non-adherence for at least two times/week has still proven to be intentional in around 25% of subjects [41], various support strategies based on novel technologies and electronic devices have been introduced over the past few years, with the aim of promoting adherence in adolescents with asthma [42]. Nevertheless, a review dedicated to these new trends showed that the adolescents’ adherence to inhalers recorded by electronic monitoring was at less than 50% [43], even if recent experiences with electronic reminders are showing encouraging results from this point of view [44,45]. However, although previous studies investigating these interventions, aiming to promote adherence in adolescents with asthma, proved to be of limited effectiveness [46,47,48], additional studies oriented to better understanding and possibly removing the deeper reasons for such poor attitudes in adolescents regarding maintaining long-term asthma strategies are still needed, even in the case of mild asthma [49].

The present study has several weaknesses and strengths. The main points of weakness are that the survey is monocentric and the sample is limited. The points of strength are: the study is drawn from a longitudinal survey, which was conducted using a large database via Boolean selection; the data of all variables were derived from the database according to the predefined times of the survey; adherence to treatment was directly assessed on a month-by-month basis for twelve months; the sub-groups of adherent and non-adherent asthma adolescents had the same consistency and were well matched; the trends of all outcomes were compared using trend analysis. 

## 5. Conclusions 

Long-term adherence to inhalation treatments is largely sub-optimal in adolescents with mild to moderate asthma, even if this is limited to a once-daily strategy. Good adherence to regular long-acting ICS-LABA inhalation allows the persistent normalization of lung function, bronchial hyperreactivity, and the optimization of health outcomes in the vast majority of cases, independently of age and gender. 

The long-term outcomes achievable in real life by good adherence to treatment should be publicized among adolescents with asthma in order to support their acceptance of the therapeutical interventions prescribed, to provide positive evidence that proper compliance can lead to substantial improvements in their quality of life, and to enhance their confidence regarding asthma. All actions oriented in this direction by families, educators, and health professionals should be extensively promoted because of their great value for young patients and society in general.

## Figures and Tables

**Figure 1 children-09-01854-f001:**
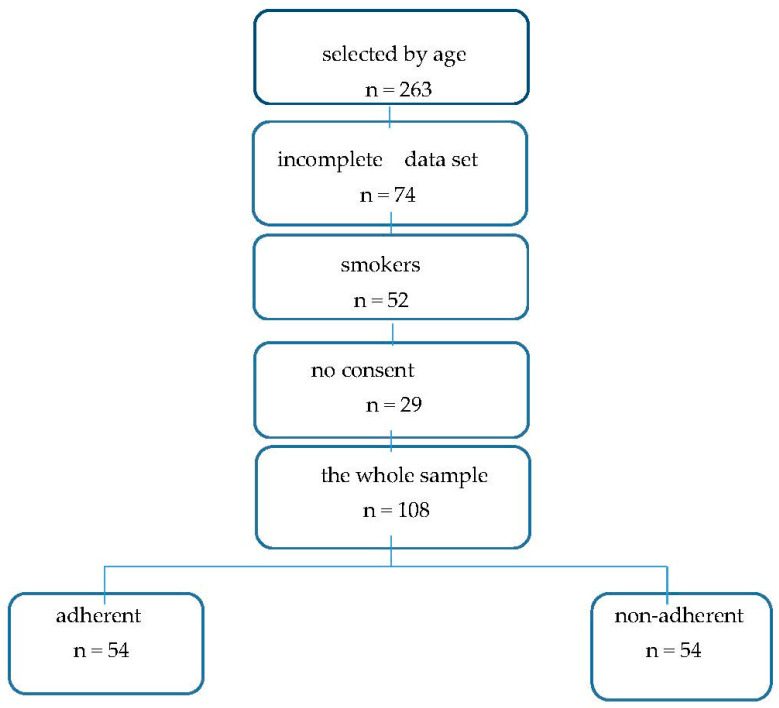
The consistency of the sample.

**Table 1 children-09-01854-t001:** General data of the whole sample, together with the mean values of lung function. Parameters measured at recruitment and health outcomes were assessed over the previous six months. (*n*; %; means ± sd).

Subjects (*n*)	108
males/females (*n*)	52/56
mean age	15.5 ± 1.8
BMI	21.8 ± 0.6
Allegens (*n*)	
only seasonal	22
seasonal & perennial	61
only perennial	25
**Lung function at recruitment**	
FEV^1^ (% pred.)	85.8 ± 14.7
MMEF (% pred.)	52.8 ± 18.7
V25 (% pred.)	45.2 ± 19.1
PD20 FEV^1^ (mcg)	877.4 ± 526.9
**Outcomes over the 6 months before recruitment**	
Exacerbations (*n*)	0.9 ± 0.9
School days off (*n*)	2.6 ± 2.3
GP visits (*n*)	1.4 ± 1.4
Specialist visits (*n*)	1.6 ± 1.2
Steroid courses (*n*)	0.9 ± 0.3
Antibiotic courses (*n*)	0.9 ± 1.0

**Table 2 children-09-01854-t002:** Mean values (means ± sd) of lung function parameters assessed at recruitment, and after 6 and 12 months of the study, in compliant and non-compliant adolescents. Statistical comparisons (ANOVA) and corresponding levels of significance are shown.

	Compliant				Non-Compliant			
	At Recruitment	+6 Months	+12 Months	*p*	At Recruitment	6 Months	12 Months	*p*
FEV^1^ (% pred.)	85.2 ± 15.5	91.8 ± 17.1	93.7 ± 18.6	0.002	87.4 ± 14.1	88.1 ± 17.2	87.6 ± 204.4	ns
MMEF (% pred.)	51.4 ± 17.3	55.8 ± 16.1	57.3 ± 17.4	0.001	52.9 ± 19.4	50.1 ± 20.6	48.7 ± 23.7	ns
V25 (% pred.)	44.8 ± 18.1	48.9 ± 18.2	50.7 ± 17.3	0.001	45.1 ± 19.5	39.9 ± 17.9	40.8 ± 20.7	ns
PD20 FEV^1^ (mcg)	807.1 ± 411.4	1289.7 ± 521.2	1408.6 ± 618.2	0.001	908.7 ± 544.1	894.2 ± 621.3	801.6 ± 704.2	ns

**Table 3 children-09-01854-t003:** Mean values (means ± sd) of health outcomes assessed over the six months before recruitment, and over 6 and 12 months of the study in compliant and non-compliant adolescents. Statistical comparisons (ANOVA) and corresponding levels of significance.

	Compliant				Non-Compliant			
	Over 6 Months before Recruitment	+6 Months	+12 Months	*p*	Over 6 Months before Recruitment	+6 Months	+12 Months	*p*
Exacerbations (*n*)	0.9 ± 0.8	0.5 ± 0.7	0.3 ± 0.6	<0.002	0.8 ± 0.9	0.8 ± 0.9	0.9 ± 1.0	ns
School days off (*n*)	2.5 ± 2.5	0.4 ± 0.8	0.2 ± 0.8	<0.001	2.7 ± 2.2	2.9 ± 2.3	2.7 ± 2.3	ns
GP visits (*n*)	1.6 ± 1.8	0.5 ± 0.8	0.2 ± 0.4	<0.001	1.5 ± 1.1	1.7 ± 1.6	1.6 ± 1.6	ns
Specialist visits (*n*)	1.5 ± 1.3	0.9 ± 0.9	0.8 ± 0.7	<0.002	1.7 ± 1.1	1.7 ± 1.2	1.7 ± 1.4	ns
Steroid courses (*n*)	0.9 ± 0.8	0.2 ± 0.4	0.1 ± 0.3	<0.001	0.8 ± 0.9	0.7 ± 0.87	0.7 ± 1.0	ns
Antibiotic courses (*n*)	0.9 ± 1.1	0.4 ± 0.7	0.1 ± 0.3	<0.001	0.9 ± 0.9	0.9 ± 0.9	0.8 ± 0.8	ns

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
