# Peer review of "Effects of Adherence to Once-Daily Treatment on Lung Function, Bronchial Hyperreactivity and Health Outcomes in Adolescents with Mild-to-Moderate Asthmoka: A Twelve-Month Survey"

_children, 2022, doi:10.3390/children9121854_

Round 1
Reviewer 1 Report
Dear colleagues, thank you for the excellent paper submitted. I had the opportunity to read your work dedicated to a major issue in asthma and other chronic diseases long term treatment. Please allow me to give you my sincere congratulations for your work.
Marginal issues - Page 3
row 92-96 In several real life studies it has been documented that some patients have the tendency to exhibit a dissimulating behavior with a deliberate purpose of gaining physician's confidence or other potential benefits. Some of them discharge many actuations from device just to have the correct numbers on the counter, before visit, in spite of being non-compliant.
Numbers collected were just stated by interviewed person or another approach was used?
row 99-102 regarding non-adherent adolescents. What approach has been done after first evaluation at 6 months with these patients, in order to regain adherence? Please elaborate. They were just observed in study or a certain retraining or other dedicated approaches were used? Did you took a non-interference approach to this group? If yes, this was assumed a-priori and included in the ethic committee discussion, before study?
I am asking because of slight [statistically not significant] deterioration of lung function observed [PD20 increased, MMEF and V25 decrease] that could generate further patterns of disease and behavior.
page 7
row 221-229 among factors influencing non-adherence in this age group that starts travelling because of school or academic-exchange programs should be included also substantial differences in management and diagnostic approaches around Europe as proved by a recent paper [Santos-Valente E, et al - Biologicals in childhood severe asthma: the European PERMEABLE survey on the status quo. ERJ Open Res. 2021 Aug 16;7(3):00143-2021. doi: 10.1183/23120541.00143-2021. PMID: 34409097; PMCID: PMC8365152.]
page 8
row 267-271 recent data have been proving that electronic reminders and incentives via a mobile phone app [again mobile phone, this essential "survival tool" for almost all teenagers...] coupled with electronic monitoring devices are feasible and acceptable to adolescents with asthma. [De Simoni A, Fleming L, Holliday L, Horne R, Priebe S, Bush A, Sheikh A, Griffiths C. Electronic reminders and rewards to improve adherence to inhaled asthma treatment in adolescents: a non-randomised feasibility study in tertiary care. BMJ Open. 2021 Oct 29;11(10):e053268. doi: 10.1136/bmjopen-2021-053268. PMID: 34716166; PMCID: PMC8559117]
there are also data that prove that it is feasible to recruit and retain young adults to examine efficacy and effectiveness and that young adults living with asthma may find certain apps to be usable, acceptable, and feasible to support adherence to ICS. [Murphy J, McSharry J, Hynes L, Molloy GJ. A Smartphone App to Support Adherence to Inhaled Corticosteroids in Young Adults With Asthma: Multi-Methods Feasibility Study. JMIR Form Res. 2021 Sep 1;5(9):e28784. doi: 10.2196/28784. PMID: 34468325; PMCID: PMC8444040.]
And other procedures that are subject of non-adherence in this age group could be addressed by electronic reminders or and e-feedback [Timm LH, Farrag G, Wolf D, Baxmann M, Schwendicke F. Effect of electronic reminders on patients' compliance during clear aligner treatment: an interrupted time series study. Sci Rep. 2022 Oct 5;12(1):16652. doi: 10.1038/s41598-022-20820-5. PMID: 36198717; PMCID: PMC9534859.]
Forgetting to take preventer inhalers was a recognized common factor affecting adherence [De Simoni A, Horne R, Fleming L, Bush A, Griffiths C. What do adolescents with asthma really think about adherence to inhalers? Insights from a qualitative analysis of a UK online forum. BMJ Open. 2017 Jun 13;7(6):e015245. doi: 10.1136/bmjopen-2016-015245. PMID: 28615272; PMCID: PMC5734261.] and e-reminders or mobile phones apps could improve this aspect in the new post-pandemic landscape of communication that has implemented new behaviors.
Author Response
First of all, I would like to thank very much the Reviewer for defining "excellent" the paper submitted, anf for his/her congratulations for our work. Thankyou very much indeed.
Marginal issue:
Pag. 3:
- row 92-96: I agree with you and we also thought to this aspect when planning the study. As the direct check was impossible for each adolescent, we decided his way although aware of possible biases. On the other hand, this kind of bias would be occur in all subjects, but outcomes proved quite poor only in those claiming <70% of doses assumed, regardless if the adolescent of one of his/her parent communicated the residual n. of actuations in the device. The overall results tend to support the model.
- row 99-102: as at the beginning of the study, also at the end of the first semester, all subjects were equally sensibilized once again to the relevance of regular daily therapy for asthma control. The role of deliberate non adherence is likely crucial in some subjects. Lung function and PD20 tended to decrease or to remain stable over time, but not to improve, in non adherent subjects.
Pag. 7:
- row 221-229: the reference you indicated has been addedd (in red)
Pag. 8
- row 267-271: thankyou for your comments: the text and References were implemented (in red) according to your suggestions.

Reviewer 2 Report
The paper "Trend of lung function, bronchial hyperreactivity, and health outcomes in adolescents with mild-to-moderate asthma by their adherence to once-daily treatment. A twelve-month survey" by Dal Negro R.W et al approaches an important clinical aspect in pulmonary medicine, compliance/adherence in asthma in young people.
There are some aspects that can be improved.
You analysed a particular ICS-LABA combination with a particular device. In think you should comment about the differences between devices, or you should name the combination in the title, including the device.
line 40. reference 8 is incomplete in the reference list. The complete citation is: Dut R, Soyer O, Sahiner UM, Esenboga S, Gur Cetinkaya P, Akgul S, Derman O, Sekerel BE, Kanbur N. Psychological burden of asthma in adolescents and their parents. J Asthma. 2022 Jun;59(6):1116-1121. doi: 10.1080/02770903.2021.1903916. Epub 2021 Mar 27. PMID: 33722151.
Line 166. Can you add data about the degree in non-adherence, in percentages?
Line 225- "similar data were recently confirmed" and you cite a paper from 2005.
Line 253. The paragraph needs editing.
The main limitation is the small number of patients.
Author Response
- as reported in Methods, fluticasone fumarate/vilanterol 90/22mcg dry powder o.d. was the combination. The name of the device (Ellipta) was added (in red).
- line 40: the refrence n. 8 has been updated and implemented (in red)
- line 225: "recently" was removed.
- line 253: the paragraph has been reworded.
- in the last paragraph of discussion we had already mentioned the limited sample of subjects.,

Reviewer 3 Report
First of all, I would like to congratulate the authors for the presentation of the article, for its relevance and transcendence.
I believe that some aspects of the article should be improved. In the introduction, it would be necessary to go more deeply into psychosocial aspects and not just indicate that they are difficult to analyze. The relationship between psychosocial factors and health is indisputable, although they are obviously difficult to investigate as is well indicated in the text. I think this is the main reason for the weakness of the article. Even more so when it is assessing, in essence, attitudes and elements linked to parenting and supervision and care of children.
It would be interesting to frame the present study in a context with multiple variables to give it relevance and not confuse. Otherwise, I would reformulate the article by simplifying the object of study, the development of the article being coherent with this object.
I would put the limitations in a separate section to give it more relevance as well as the ethical considerations. In addition, I would complete the key words. As for the instrument used, it is timely if the psychosocial variables are not taken into account.
Best regards codiales.
Author Response
Thankyou for your comment.
Anyway, Introduction reports more than 50 lines in mentioning main different causes of adolescents' lower adherence to respiratory treatment , and the 90% were devoted to related psychological aspects. The study was not aimed to investigate specifically all aspects of adolescents' psychological determinants of their low adherence. Different special skills are required.
Moreover, the aim of the present study was to assess how much different degrees of adherence to prescribed medications, whatever the main psychological cause, can affect lung function and health outcomes over time: this was the main object of the study.

Round 2
Reviewer 3 Report
I understand the object of study of the article but I think it is a biased way of approaching the knowledge of the problem, even though the introduction talks about it. However, as it is more or less clear in the text, I am satisfied.
The key words are still incomplete. Only asthma...
Best regards